# Guided anisotropic oxygen transport in vacancy ordered oxides

Zhenzhong Yang [1,2,7], Le Wang [1,7], Jeffrey A. Dhas[3,6], Mark H. Engelhard [4], Mark E. Bowden [4], Wen Liu [4,5], Zihua Zhu [4], Chongmin Wang [4], Scott A. Chambers [1], Peter V. Sushko [1] ✉ & Yingge Du [1] ✉

Anisotropic and efficient transport of ions under external stimuli governs the operation and failure mechanisms of energy-conversion systems and micro-electronics devices. However, fundamental understanding of ion hopping processes is impeded by the lack of atomically precise materials and probes that allow for the monitoring and control at the appropriate time- and length-scales. In this work, using in-situ transmission electron microscopy, we directly show that oxygen ion migration in vacancy ordered, semiconducting $SrFeO_{2.5}$ epitaxial thin films can be guided to proceed through two distinctly different diffusion pathways, each resulting in different polymorphs of $SrFeO_{2.75}$ with different ground electronic properties before reaching a fully oxidized, metallic $SrFeO_3$ phase. The diffusion steps and reaction intermediates are revealed by means of ab-initio calculations. The principles of controlling oxygen diffusion pathways and reaction intermediates demonstrated here may advance the rational design of structurally ordered oxides for tailored applications and provide insights for developing devices with multiple states of regulation.

$ABO_3$-type perovskite-structured transition metal oxides and their structural variances (e.g., Ruddlesden-Popper, Brownmillerite, infinite-layer phases) have been extensively studied because of their remarkable physicochemical properties, including metal-to-insulator transition (MIT)[1-4], superconductivity[5,6], ferroelectricity[7,8], notably high ionic conduction[9,10], and surface catalytic activity[11-14]. Oxygen-based defects, which may be present as isolated oxygen vacancies ($V_O$), vacancy clusters, or ordered oxygen vacancy channels (OVCs), can affect and even dominate these properties[4,15-19]. The topotactic phase transitions (TPT) as a result of oxygen content change can lead to intriguing concurrent changes in electronic, optical, and magnetic properties[16,18,19]. Memristors, synaptic transistors, and high-density memories based on such TPTs have been designed and tested, offering

significant promise in the future oxide electronics and neuromorphic computing[4,20,21]. Reliability of the existing types of devices and our ability to create advanced and robust signal processing systems are predicated on the understanding of the coupling between composition, structure, electronic properties, and external stimuli.

Strontium ferrite ($SrFeO_x$) is a convenient platform to study reversible redox activity and associated property changes. Oxygen-deficient brownmillerite-structured $SrFeO_{2.5}$ (BM-SFO) (Fig. 1a, c), with ordered oxygen vacancy channels, can exhibit facile, highly anisotropic oxygen ion transport and low energy diffusion barriers within the OVCs, making these materials attractive for use as electrocatalysts and oxygen membranes[16,17]. BM-SFO is a G-type antiferromagnetic insulator, whereas the stoichiometric perovskite-structured $SrFeO_3$

[1]Physical and Computational Sciences Directorate, Pacific Northwest National Laboratory, Richland, WA 99354, USA. [2]Key Laboratory of Polar Materials and Devices (MOE), Department of Electronics, East China Normal University, Shanghai 200241, P. R. China. [3]School of Chemical, Biological and Environmental Engineering, Oregon State University, Corvallis, OR 97331, USA. [4]Environmental Molecular Sciences Laboratory, Pacific Northwest National Laboratory, Richland, WA 99354, USA. [5]State Key Laboratory of Biogeology and Environmental Geology, China University of Geosciences, Wuhan 430074, China. [6]Present address: Physical and Computational Sciences Directorate, Pacific Northwest National Laboratory, Richland, WA 99354, USA. [7]These authors contributed equally: Zhenzhong Yang, Le Wang. ✉e-mail: peter.sushko@pnnl.gov; yingge.du@pnnl.gov

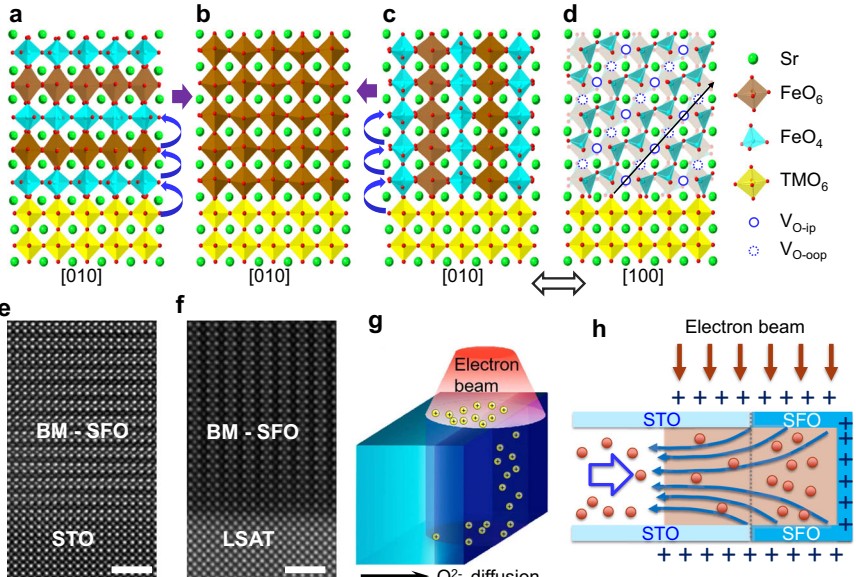

**Fig. 1 | Characterization of BM-SFO thin films and the concept of oxidation of BM-SFO by in-situ TEM. a–c** Structure models (projected along [010]) of (**a**) BM-SFO with OVCs parallel to the substrate, (**b**) P-SFO, and (**c**) BM-SFO with OVCs perpendicular to substrate. **d** Structure model highlighting an OVC plane shown in (**c**) but viewed along [100]. **e–f** HAADF-STEM images of BM-SFO grown on STO(001) and LSAT(001), respectively, demonstrating the controlled orientation of OVCs. Scale bar is 2 nm. **g** Schematic illustrations showing the electron-beam illumination of the cross-sectional STO/SFO sample. The yellow spheres represent the positively charged surface of the specimen as a result of electron beam illumination. **h** Schematics of electron-beam induced electrical field (dark blue arrows) generated during TEM imaging and its effect in promoting $O^{2-}$ diffusion. The orange spheres represent $O^{2-}$ ions moving in the opposite direction of electron-beam-induced electrical field. The light brown color represents the area under electron beam illumination.

(P-SFO) (Fig. 1b) is an antiferromagnetic metal[1,4,15], which suggests that a zoo of transitory electronic behaviors may be realized by accessing metastable phases along the BM-SFO → P-SFO transition pathways. For example, memristive switching behavior is attributed to the dynamic formation/breaking of a conductive P-SFO filament in the parent matrix of insulating BM-SFO[22,23]. However, an atomistic understanding that bridges the evolution of the microstructure and properties in such materials is still lacking, preventing their predictive and practical use.

In this work, we show that controlling the orientation of the OVCs by epitaxial growth allows us to exploit the structural anisotropy to stabilize isomeric yet structurally and electronically distinct intermediates (SrFeO$_{2.75}$). We use in-situ transmission electron microscopy (TEM) to directly activate and image the TPT from BM-SFO to P-SFO, where oxygen ions are supplied from the reducible substrates under a built-up electric field created by electron beam irradiation during TEM imaging. Oxygen diffusion pathways that allow us to selectively access metastable reaction intermediate phases are revealed using density functional theory (DFT) calculations. Our results provide atomic-scale insights into oxygen diffusion and redox-driven phase transition processes occurring in vacancy-ordered oxides, paving the way for a deliberate control of the metastable phases.

## Results

### Synthesis and characterization of BM-SFO thin films

Figure 1a, c show the structure model (viewed along [010]) of BM-SFO with two orientations of OVCs. BM-SFO is composed of alternating oxygen-deficient FeO$_4$ tetrahedral layers (i.e., OVCs) and fully coordinated FeO$_6$ octahedral layers. If OVCs are aligned parallel to the interface (Fig. 1a), oxygen ion ($O^{2-}$) diffusion along the out-of-plane direction (i.e., from octahedral FeO$_6$ sublayers to tetrahedral FeO$_4$ sublayers) would proceed with a higher energy barrier[21,24]. In contrast, if OVCs are perpendicular to the interface (Fig. 1c, d), $O^{2-}$ migrating from the substrate can directly intercalate into OVCs without crossing the FeO$_6$ layers and, therefore, with a lower energy barrier. We note that manipulating the orientation of OVCs has been realized in a

number of systems (such as cobaltites[25–27] and ferrites[10,19,28–30]) over the years. In addition to conventionally used controls of the thin film deposition parameters and substrates, Han et al[31] recently reported on controlling the OVC ordering in SrCoO$_{3-\delta}$ thin layers via ionic liquid gating (ILG), which may open a path toward the creation of oxitronic devices.

Here, we choose BM-SFO thin films with differently oriented OVCs as our studied materials to probe the effect of OVCs on anisotropic $O^{2-}$ migration. Epitaxial BM-SFO thin films with the thicknesses of 15–35 nm were deposited on (001)-oriented SrTiO$_3$ (STO) and (LaAlO$_3$)$_{0.3}$(Sr$_2$AlTaO$_6$)$_{0.7}$ (LSAT) single crystal substrates by using pulsed laser deposition (PLD). STO ($a = 3.905$ Å) and LSAT ($a = 3.868$ Å) substrates were used because of their small in-plane lattice mismatches with BM-SFO and their reducibility, which can serve as an oxygen source/vacancy sink[32–34]. XRD $\theta - 2\theta$ scans along the out-of-plane direction (Supplementary Fig. 1) demonstrate that OVCs of BM-SFO can be stabilized to be either parallel to the interface with STO or perpendicular to the interface with LSAT. Moreover, high-angle annular dark-field scanning TEM (HAADF-STEM) images (Fig. 1e, f) further verify the characteristic of vacancy ordering in BM-SFO, where the OVCs appear as dark stripes repeating in alternate layer[15,19,28,35]. High-quality BM-SFO thin films with different orientations of OVCs enable us to further investigate the details of anisotropic oxygen transport in these materials by in-situ TEM.

### Oxidation of BM-SFO thin films by in-situ TEM

In-situ TEM is used to drive the oxidation of BM-SFO and monitor subsequent phase transitions. Previous studies have shown that electron beam illumination on an insulating sample during TEM imaging can generate secondary and Auger electrons that subsequently leave the sample surface[36–39]. The outer shell of the sample can become positively charged due to the loss of these electrons and an additional electric field is generated (Fig. 1g–h and Supplementary Fig. 2). It has been reported that a few volts can be induced at an electron beam dose of $10^3$–$10^4$ e Å$^{-2}$ s$^{-1}$ [40], which could be sufficient to drive the O species

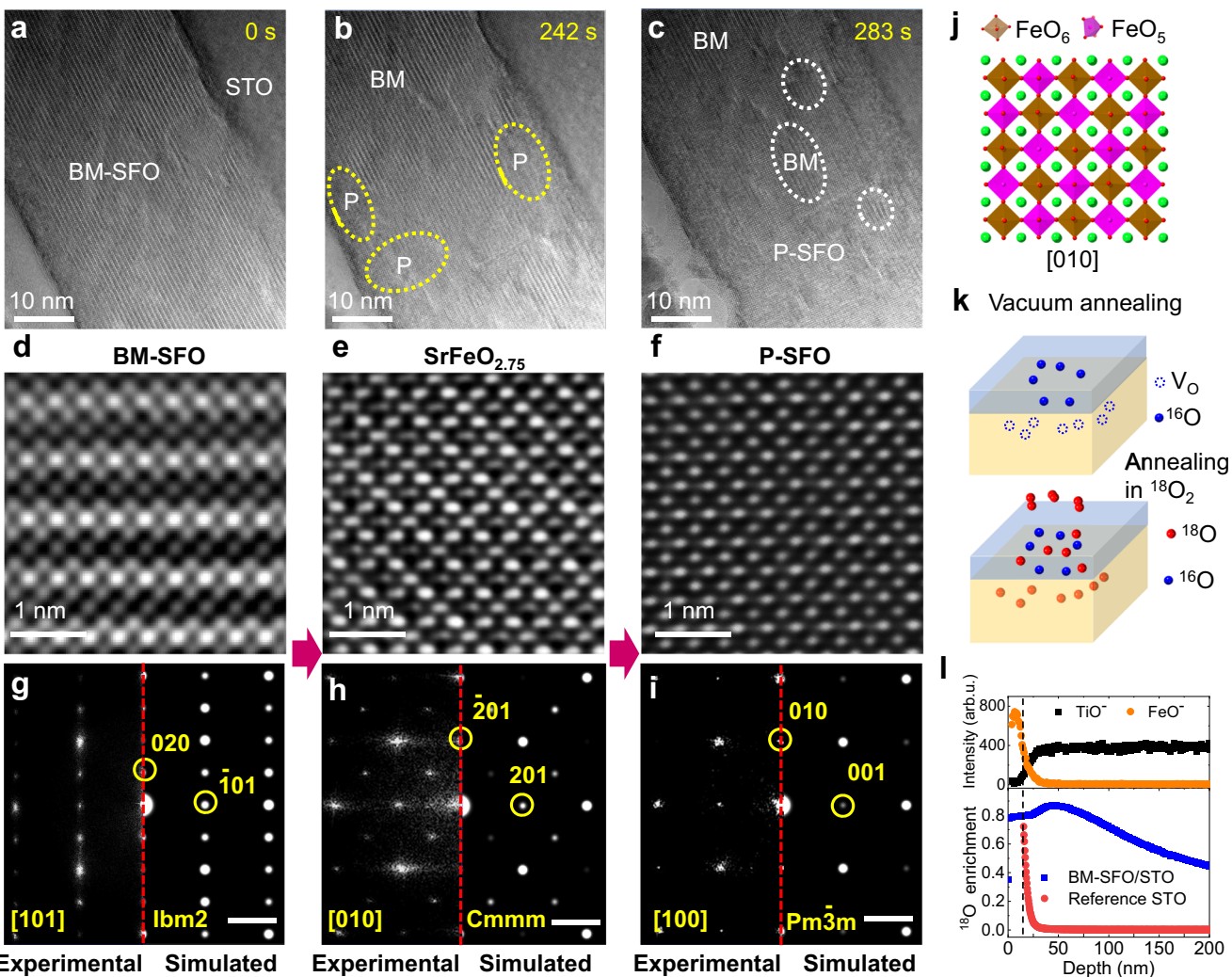

**Fig. 2 | Phase transformations of BM-SFO → SrFeO$_{2.75}$ → P-SFO observed on BM-SFO/STO with starting OVCs parallel to STO(001). a–c** Selected in-situ TEM images (see Supplementary Movie 1) of the same region of a ≈35 nm SFO film illustrating stages of the phase transition process over ≈300 s: the initial state of BM-SFO (**a**), appearance of the P-SFO inclusions (**b**), and propagation of the P-SFO phase (from the bottom of the image) (**c**). **d–f** High resolution TEM images and **g–i**, Fast Fourier Transform patterns (bottom left) and simulated diffraction patterns (bottom right) for (**d**) BM-SFO, (**e**) SrFeO$_{2.75}$, and (**f**) P-SFO. The scale bars are 2 nm⁻¹. **j** Structure model of bulk-like SrFeO$_{2.75}$. **k** Schematics showing the creation of oxygen vacancies (V$_O$s) in SFO/STO by annealing in vacuum, and subsequent replenishment of oxygen by annealing in ¹⁸O$_2$. **l** ToF-SIMS depth profiles of ¹⁸O display the enrichment level for a 15 nm SFO film grown on STO(001) and a STO reference sample. The SFO/STO interface location, marked by the black dashed line, was confirmed by the secondary ion signals of FeO⁻ and TiO⁻.

migration from the reducible substrate STO or LSAT to BM-SFO[15,21,40,41]. Furthermore, the samples are heated to elevated temperatures (200 to 300 °C) to lower the O migration energy barrier in our setup during in-situ TEM experiments, allowing us to further tune the reaction speed so that the structural evolution from BM-SFO to P-SFO can be monitored at an appropriate time scale. It is worth noting that, in our experiment, neither electron beam illumination nor the elevated temperature heating could work alone to induce adequate oxygen ions flux to oxidize the BM-SFO under vacuum, which is consistent with former results[1,42]. While combining the electron beam-induced electric field and low-temperature thermal field, the adequate O ions migration from STO to SFO could be triggered and thus realized the oxidation process of SFO.

### Oxygen ion diffusion across the OVC layers

In-situ TEM studies on BM-SFO/STO, in which OVCs are parallel to the interface (Fig. 1a, e), allowed us to capture the dynamical phase transition processes (Supplementary Movie 1, recorded under 300 °C sample heating). We note that the rate of O accumulation in SFO is determined by two processes: O ions migration from STO to SFO

during the in-situ TEM experiments and O loss from SFO thin film that occurs at 300 °C under vacuum. The loss of O from SFO reduces the effective oxygen flux in SFO and, therefore, decreases the P-SFO nucleation rate. It took almost 4 minutes for the nucleation of the P-SFO phase to take place. Figure 2a–c show snapshot TEM images taken from Supplementary Movie 1, demonstrating the formation and evolution of the P-SFO phase in the BM-SFO matrix. The high-resolution TEM images corresponding to the starting BM-SFO and emerging P-SFO are shown in the panels of Fig. 2d, f, respectively. As shown in the bottom panels of Fig. 2g, i, the Fast Fourier Transform (FFT) patterns obtained from the TEM images match well with the simulated diffraction patterns that were based on the crystal structures of the corresponding phases. Furthermore, HAADF and annular bright-field (ABF)-STEM imaging for BM-SFO and P-SFO phases are also conducted, shown in Supplementary Fig. 3, to demonstrate the details of structure change at atomic scale.

It is worth to note that the formed P-SFO phase can quickly transform back to BM-SFO if the electron beam is blanked as we conduct the in-situ TEM experiment at elevated temperature (300 °C), suggesting rapid O emission from P-SFO due to vacuum annealing. We

used a lower heating temperature of 200 °C during another in-situ TEM experiment to slow down the O loss process, which also allowed us to capture an intermediate phase displaying a checkerboard-like structure in the transition regions (see Supplementary Movie 2 and Fig. 2e). In this case, it took about 1 min and a half for the starting of phase transition, which is much shorter than the waiting time for P-SFO nucleation under the heating temperature of 300 °C, suggesting the much slower O emission from SFO thin film under 200 °C. The intermediate phase's FFT pattern well matches with the simulated diffraction pattern of a previously observed $SrFeO_{2.75}$ phase (Cmmm space group, Fig. 2h)[43,44]. The structure of the $SrFeO_{2.75}$ phase (Fig. 2e, j) shows a checkerboard pattern composed of alternating columns of octahedral $FeO_6$ and pyramidal $FeO_5$. This ordering corresponds to the formation of oxygen-deficient $FeO_5$ connectivity (chain of $V_{Os}$) along the beam projection direction.

As the experiments were conducted in ultra-high vacuum, there is no other oxygen source but the STO substrates. In addition, according to the built-up electrical field shown in Fig. 1g, we expect the oxygen ions needed to heal the $V_{Os}$ to form $SrFeO_{2.75}$ or P-SFO should come from the reducible STO substrate, resulting in local oxygen deficiency at the vicinity of film/STO interface. To further confirm this, we collected cross-sectional STEM-electron energy loss spectroscopy (EELS) maps at the SFO/STO heterostructure at room temperature after in-situ TEM experiments. As shown in Supplementary Fig. 4, the O intensity for STO exhibits a visible decrease in the region close to the SFO/STO interface (≈30 nm in width), supporting our hypothesis. The emergence of clusters-like contrast (indicated by circles in Supplementary Fig. 4) close to the SFO/STO interface also indicates the agglomeration of $V_{Os}$[45,46].

To further visualize the oxygen exchange between SFO and STO, we performed $^{18}O$ labeled, time-of-flight secondary-ion mass spectrometry (ToF-SIMS) analysis. One BM-SFO/STO sample (≈15 nm thickness) and a STO(001) substrate were first annealed in vacuum at 700 °C for 1 h. For BM-SFO/STO sample, vacuum annealing (VA) could promote oxygen loss in SFO, which would subsequently draw oxygen from the STO substrate. This is similar to the well-established phenomenon of oxygen transfer from the substrate to the film deposited by PLD[32,33]. After VA, the BM-SFO/STO sample and the STO substrate (STO reference marked in Fig. 2l) were annealed at 650 °C for 0.5 h in a tube furnace backfilled with 50 Torr of $^{18}O_2$.

To account for the possibility of matrix effects during ToF-SIMS analysis, one non-$^{18}O_2$ annealed BM-SFO/STO sample was measured (Supplementary Fig. 5). The oxygen signal intensity shows a notable increase upon sputtering through the BM-SFO film into the STO substrate (Supplementary Fig. 5b). Using the ratio of the total oxygen signal intensity on the film and substrate sides (i.e., $(^{16}O + {}^{18}O)_{film}/(^{16}O + {}^{18}O)_{substrate}$) yields a ratio of 0.80, which is close to the expected stoichiometric ratio of oxygen in $BM-SrFeO_{2.5}/SrTiO_3$ of 0.83. Therefore, this result suggests that the matrix effect, which has been well described in previous reports[47–50], does not play a dominant role in the oxygen signal intensity change upon sputtering through the BM-SFO film into the STO substrate.

Based on prior reports[47–50], matrix effects can significantly influence signal intensity but play a minimal role in isotopic ratio counting. Indeed, as demonstrated in Supplementary Fig. 5, the ratio of $^{18}O/(^{16}O + {}^{18}O)$ for a non-$^{18}O_2$ annealed BM-SFO/STO remains constant at the natural abundance of $^{18}O$ (Supplementary Fig. 5c), even though the signal intensity changes from BM-SFO to STO. Hence, we utilized the $^{18}O$ enrichment level, defined as $^{18}O/(^{18}O + {}^{16}O)$, to assess and visualize the oxygen exchange occurring at the interface. Figure 2l presents the depth profiles obtained from ToF-SIMS, illustrating the $^{18}O$ enrichment level for both $^{18}O_2$ annealed samples.

The $^{18}O$ enrichment level is ≈74% at the near-surface region of STO reference, which undergoes an exponential decay to natural abundance level (≈0.2%) within a ≈100 nm range (see Supplementary Fig. 6).

In comparison, the $^{18}O$ enrichment level is higher (≈80% in SFO film) in the BM-SFO/STO sample due to the strong surface oxygen exchange between $^{18}O_2$ and SFO. An even higher $^{18}O$ enrichment level (up to ≈87%) is also observed on the STO side close to the STO/SFO interface, indicating that the oxygen vacancies in the STO substrate created during the VA process have been filled by $^{18}O_2$ annealing (see schematics shown in Fig. 2k). We also performed $^{18}O$ annealing experiment on an as-received STO substrate and observed similar $^{18}O$ concentration distribution with that of VA STO substrate (see Supplementary Fig. 6). These results clearly demonstrated that oxygen exchange of the STO substrate can occur at 650 °C. Moreover, the oxygen diffusion coefficients of STO at 650 °C was determined by fitting the $^{18}O$ depth profiles for both VA and non-VA STO substrates[34,51–54]. As shown in Supplementary Fig. 6, the bulk diffusion coefficient ($D^*$) for VA and non-VA STO substrates were found to be $1.2 \times 10^{-10}$ and $3.1 \times 10^{-11}$ cm² s⁻¹, respectively, in agreement with previously reported values[51,52].

## Oxygen ion diffusion within the OVCs

We now consider the case of BM-SFO/LSAT, in which OVCs are perpendicular to the interface (Fig. 1c, f). As shown in Supplementary Movies 3 and 4, compared to BM-SFO/STO, the phase transition occurred more readily (almost no waiting time) in BM-SFO/LSAT during in-situ electron beam shower and 200 °C sample heating. We attribute this to the low diffusion barrier and high migration rate of O ions along OVCs in BM-SFO. Interestingly, two reaction fronts can be observed during the phase transition process, indicating a two-step reaction during the oxidation from BM-SFO to P-SFO. In Fig. 3a–d, the sequences of TEM images taken from Supplementary Movie 3 clearly show the phase transformation from BM-SFO (Fig. 3a, e) to P-SFO (Fig. 3d, g, l). Figure 3b–c capture an intermediate phase separated by two reaction fronts (marked by white and yellow dashed lines). A magnified view of the intermediate phase circled in Fig. 3b is displayed in Fig. 3f. High-resolution HAADF-STEM image (Fig. 3h) together with its lattice spacing mapping (Fig. 3i) clearly reveal the phase boundaries, and the three phases are assigned as BM-SFO, $SrFeO_{2.75}$, and P-SFO. The experimentally measured Sr-Sr in-plane spacings for BM-SFO are ≈4.3 Å and 3.4 Å, due to the modulations from $FeO_4$ and $FeO_6$ connectivity, respectively (Fig. 3m). In contrast, the Sr-Sr distance of P-SFO is converged to one value of around 3.9 Å (Fig. 3m), as expected from a cubic structure coherently strained to an LSAT substrate. Compared to BM-SFO ($SrFeO_{2.5}$), the long and short Sr-Sr atomic distances of the intermediate phase are measured to be ≈4.0 Å and 3.8 Å, respectively (Fig. 3m). According to our DFT modeling, the intermediate $SrFeO_{2.5}$ phase has a structure that is significantly different from the bulk-$SFO_{2.75}$ phase shown in Fig. 2j. In this case, oxygen ion migration from the reducible LSAT substrate could directly intercalate into the OVCs without involving mass transfer from/to the $FeO_6$ sublayers. The $SrFeO_{2.75}$ structure contains OVCs characteristic of $FeO_5$ pyramidal connectivity (model in Fig. 3k) as a result of selectively healing $V_{Os}$. The decrease in the IP Sr-Sr atomic distance between the sub-stoichiometric layers compared to BM-SFO is expected due to partial healing of $V_{Os}$[19,55,56].

## Discussion

To gain atomic-scale insights into the electronic and structural changes upon BM-SFO oxidation and the details of different oxygen diffusion pathways related to the orientation of OVCs, we turn to DFT simulations (details are summarized in Methods). Figure 4 summarizes the relationships between oxygen content and SFO stability, oxygen diffusion pathways, and electronic properties. We first consider the case of BM-SFO/STO, in which OVCs are parallel to the interface and a bulk-like intermediate phase $SrFeO_{2.75}$ was observed upon BM-SFO oxidation. We mimic the experimentally observed O transfer from STO to $SrFeO_x$ by continuously incorporating O atoms into one OVC plane of the BM-SFO supercell. Our calculations show that such O

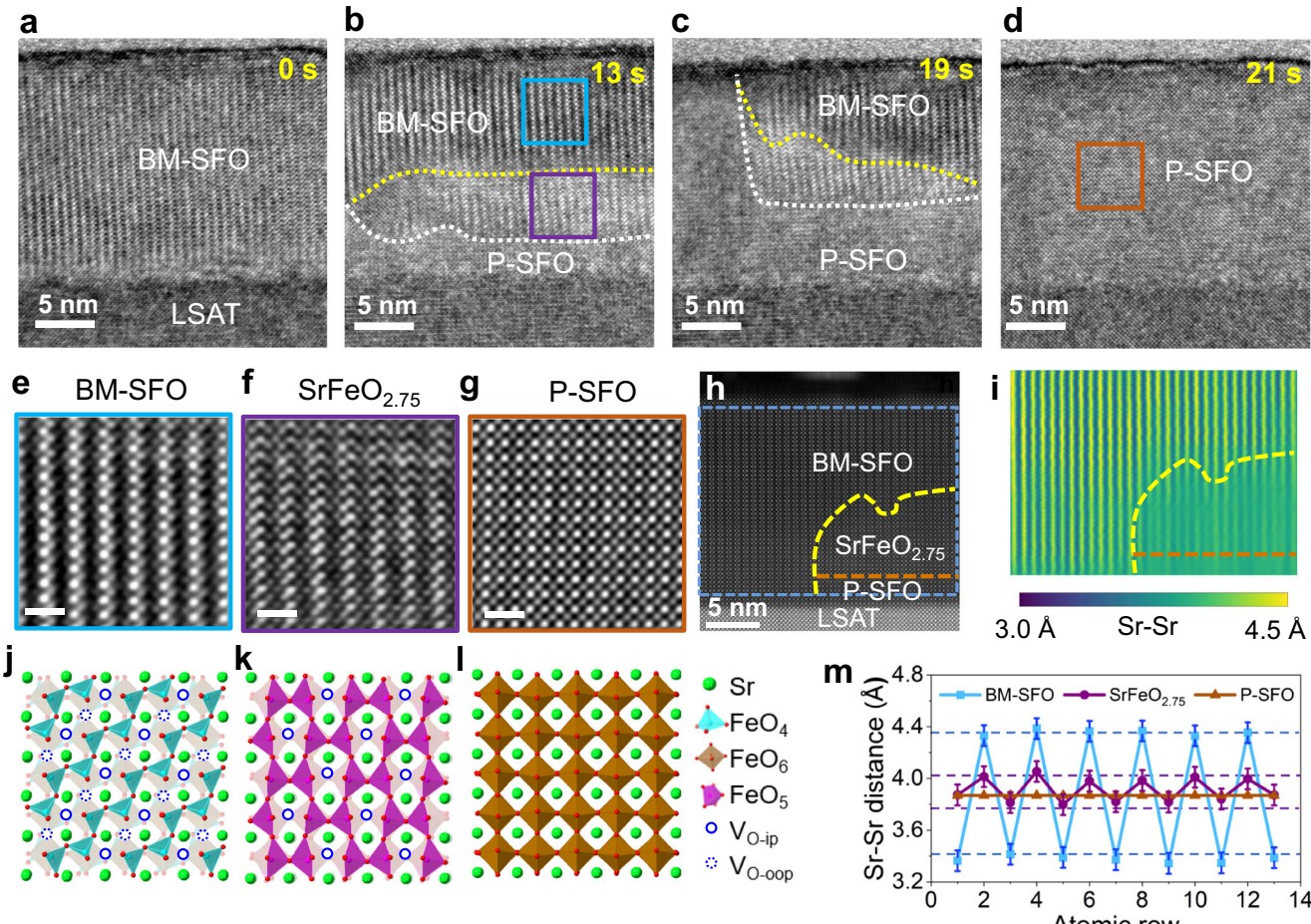

**Fig. 3 | Phase transformations of BM-SFO → SrFeO$_{2.75}$ → P-SFO observed on BM-SFO/LSAT with starting OVCs perpendicular to LSAT(001). a–d** Time-stamped in-situ TEM images taken from Supplementary Movie 3 illustrating a two-step phase transition process. The yellow and white dashed lines indicate the two reaction fronts. **e–g** High-resolution TEM images matching the highlighted regions shown in (**b**) and (**d**) and **j–l**, their corresponding structure models. **h** HAADF-STEM image of an SFO/LSAT sample showing the coexistence of three different phases. **i** In-plane lattice spacing (between Sr-Sr ions) map of the blue box marked region shown in (**h**). **m** Comparison of experimentally measured values (symbols) with DFT calculated results (dashed lines) for in-plane Sr-Sr interatomic distances among the three phases. The error bars of ≈0.1 Å is determined considering the finite pixel size of the experimental HAADF image.

accumulation is thermodynamically preferred up to $x \approx 2.7$ (Fig. 4a). These additional O species pull electron charge from the neighboring ions and, as formally O$^{2-}$ ions ($2p^6$), diffuse along the OVC parallel to the interface with STO with the calculated barrier of ≈0.65 eV (Supplementary Fig. 7). Further incorporation of O into the same OVC until all vacant sites are occupied ($x = 2.75$ for the supercell used here) is cost neutral. This configuration corresponds to the formation of a full P-SFO layer; accordingly, the P-SFO/BM-SFO interface advances to the next OVC layer. We define the SrFeO$_{2.75}$ intermediate formed through vacancy layer accumulation as LA-SFO as shown in Fig. 4a. As the concentration of the additional O species in the OVC increases, the number of vacant sites available for their diffusion decreases which slows the in-plane diffusion down. At the same time, O species in the vicinity of the P-SFO/BM-SFO interface become progressively less negative due to a competition between the preferred O$^{2-}$ and Fe$^{3+}$ electronic configurations, which leads to the formation of O$^{(2-\delta)-}$ ($2p^{6-\delta}$) and Fe$^{3+\gamma}$ ($3d^{5-\gamma}$) ions (see Supplementary Fig. 8). This depletion of the O $2p$ band destabilizes the oxygen sublattice, thus promoting local restructuring. We found that for the fixed concentration of additional oxygens ($x = 2.75$), distributing them over all OVCs, rather than confining them to one OVC is energetically preferred (Fig. 4a). The most stable configuration corresponds to the bulk SrFeO$_{2.75}$ phase (purple dot in Fig. 4a), as observed experimentally (Fig. 2e) and illustrated in Fig. 2j.

To shed light on the kinetics of the lattice reorganization, we investigated the mechanisms of oxygen ion diffusion across the FeO$_6$ layer into the next OVC in the out-of-plane direction, as schematically indicated in Fig. 1a. Our simulations suggest that this diffusion proceeds via a two-step mechanism (A–B–C path in Fig. 4b), whereby each step has the barrier of ≈0.6 eV. Notably, the transient configuration (B) formed after the first diffusion step can reverse to the original configuration (A) with the barrier of only 0.1 eV, which renders the overall diffusion barrier across the FeO$_6$ plane of ≈1.1 eV, i.e., nearly twice as large as that for diffusion within an OVC plane (≈0.65 eV). Since the formation of bulk-SFO$_{2.75}$ requires a diffusion process disrupting continuous FeO$_6$ planes by forming checkerboard arrangements of FeO$_6$ and FeO$_5$ polyhedra, we refer to it as disruptive diffusion (DD) thereafter and the corresponding bulk phase as DD-SFO$_{2.75}$ (Fig. 4a).

For the case of BM-SFO/LSAT, in which OVCs are vertically aligned to the interface and a new intermediate phase was observed upon BM-SFO oxidation. To analyze the effect of oxygen incorporation and to establish the atomic structure of this observed intermediate phase, we examined the stability of SFO depending on the arrangements of the oxygen species using DFT simulations. Figure 1d shows the side view (along [100] direction) of a vertically aligned OVC in which only connectivity of FeO$_4$ tetrahedra is visible. Healing of the in-plane V$_O$ indicated by solid red circles (defined as V$_{O-ip}$) will lead to the formation of

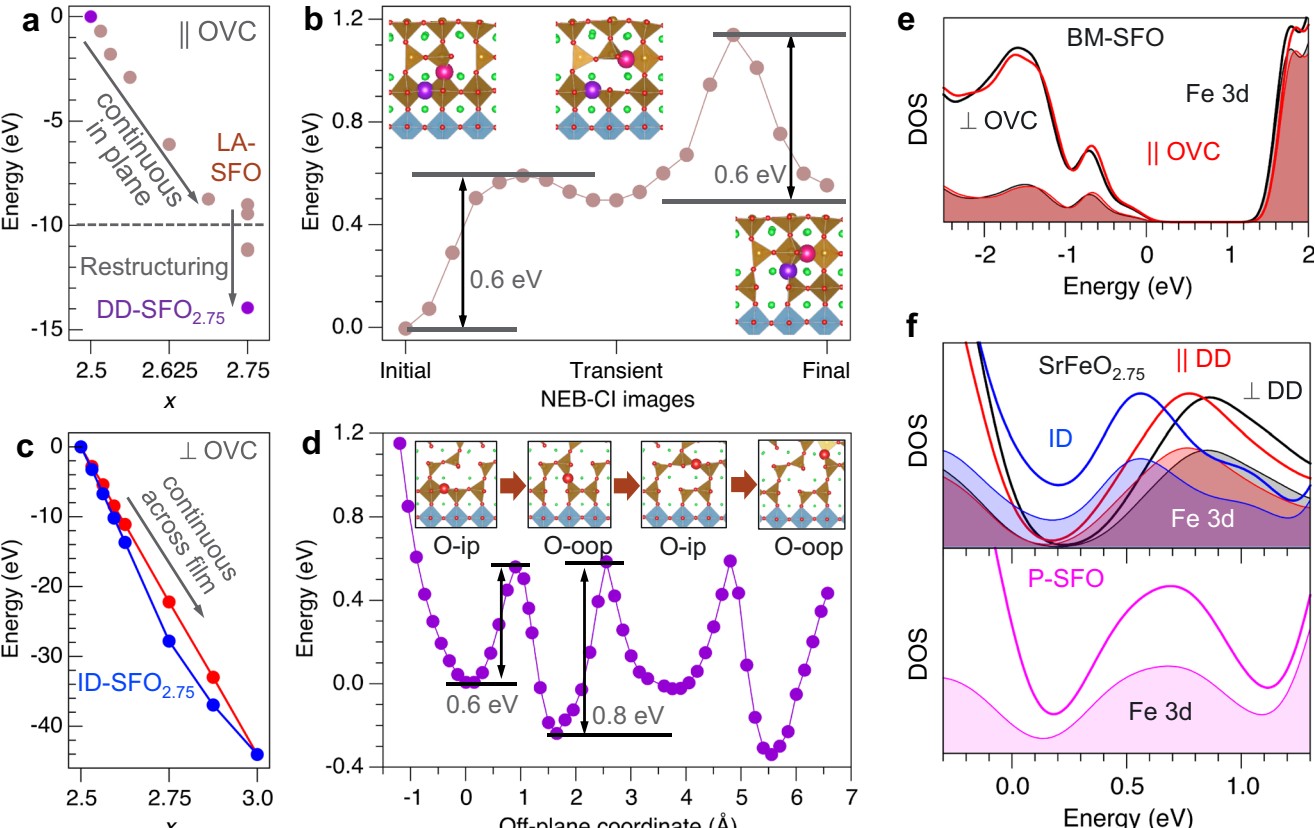

**Fig. 4 | DFT calculations of SFO stability, oxygen diffusion pathways, and electronic density of states. a** Energy gain due to incorporation of oxygen into OVCs parallel (∥) to the substrate. Incorporation proceeds continuously into one OVC layer up to ≈75% occupancy ($x ≈ 2.7$ in our supercell) with the formation of LA-SFO$_{2.75}$ phase; reorganization of the oxygen sublattice at larger $x$ through disruptive diffusion leads to the formation of DD-SFO$_{2.75}$. **b** Potential energy surface (PES) for an O$^{2-}$ (large spheres) diffusion across Fe-centered polyhedra (brown) FeO$_6$ layers into a neighboring OVC calculated using nudged elastic band climbing image (NEB-CI) method; insets show the A (initial) and C (final) configurations for each step; the transient configuration (B) in between renders the overall diffusion barrier of ≈1.1 eV. **c** Energy gain due to oxygen incorporation into OVCs perpendicular (⊥) to the substrate ($U_{eff} = 0$ eV) for sequentially occupied V$_{O-ip}$ and V$_{O-oop}$ sites (red) and sequentially occupied V$_{O-oop}$ and V$_{O-ip}$ sites (blue), leading to the formation of the intercalation ID-SFO$_{2.75}$ phase. **d** PES for O$^{2-}$ diffusion along perpendicular to OVC. Insets show the local atomic configurations for the first four minima. One-electron density of states (DOS) for **e**, BM-SFO ($x = 2.5$) and **f**, ID- and DD-SFO$_{2.75}$, and P-SFO ($x = 3$) phases. Shaded areas show DOS projected on Fe 3d states. Fermi energy is at 0 eV.

Fe-O-Fe bonds parallel to the interface. In comparison, the V$_{Os}$ formed between two Fe atoms along the out-of-plane direction (indicated by dotted circles) are defined as V$_{O-oop}$, and healing those will lead to Fe-O-Fe bonds perpendicular to the interface. Along the one-dimensional (1D) FeO$_4$ tetrahedral chain, the energies needed to heal the alternating V$_{O-ip}$ and V$_{O-oop}$ may vary depending on the sequence of oxygen incorporation and the substrate-induced strain. We considered several configurations of the O species occupying V$_{O-ip}$ only or V$_{O-oop}$ only sites and varied the concentration of these additional species (see Fig. 4c). According to our simulations for the 8 × 2 × 4 supercell (see Methods) at low O concentration, it is energetically favorable for the O to occupy V$_{O-oop}$ sites (blue symbols in Fig. 4c). The preference to occupy V$_{O-oop}$ over V$_{O-ip}$ sites persists through the entire $2.5 < x ≤ 2.75$ range. An additional oxygen in the BM-SFO lattice is more stable at the V$_{O-oop}$ site than at the V$_{O-ip}$ site by ≈0.2 eV. However, for the SrFeO$_{2.75}$ phase, this energy difference reaches 0.35 eV (for Hubbard $U = 0$ eV). Preferential occupation of the V$_{O-oop}$ sites is attributed to unconstrained lattice relaxation in the off-plane direction, while in-plane relaxation is suppressed due to epitaxial constraints. Accordingly, the out-of-plane lattice parameter decreases with increasing occupancy of the V$_{O-oop}$ sites and remains essentially unchanged with occupancy of the V$_{O-ip}$ sites. At $x = 2.75$, our DFT modeling predicts that the stable structure should have all V$_{O-oop}$ sites occupied and all V$_{O-ip}$ sites vacant (see Supplementary Fig. 9), which is selected as the structural model

for the experimentally observed reaction intermediate shown in Fig. 3f. Moreover, the long and short Sr-Sr distances (indicated by the dashed lines in Fig. 3m) in the modeled SrFeO$_{2.75}$ are calculated to be ≈4.0 Å and 3.7 Å, respectively, which are in good agreement with the experimentally measured values. Since this SrFeO$_{2.75}$ intermediate phase is derived by intercalation diffusion (ID) along the chains of adjacent V$_{O-ip}$ or V$_{O-oop}$ sites (see Fig. 1d), we define it as the ID-SFO$_{2.75}$ phase.

To obtain atomic-level insights into the kinetics of the oxidation process for the case of BM-SFO/LSAT, we determined the oxygen diffusion pathways and activation energies using DFT, in which the BM-SFO film was explicitly strained to the substrate. Figure 4d shows the potential energy surface (PES) calculated by displacing an O$^{2-}$ ion along the OVC and optimizing the positions of all other atoms at every step. The deep and shallow energy minima correspond to the O$^{2-}$ occupying V$_{O-oop}$ and V$_{O-ip}$ sites, respectively. The insets show the corresponding configurations of the selected O$^{2-}$ ion (shown in red) migrating from the in-plane to out-of-plane to in-plane configuration. The steep rise of the left side of the plot reflects the interaction of the extra oxygen in the OVC and the stoichiometric substrate. The calculated diffusion barriers vary between ≈0.6 and ≈0.8 eV, suggesting that O$^{2-}$ diffusion proceeds through the entire film with a preference of occupying the V$_{O-oop}$ sites, which explains the mechanism of the formation of the transient ID-SFO$_{2.75}$ phase observed experimentally.

The compositional and structural differences between the SFO phases are reflected in the differences of their electronic properties. Figure 4e, f shows the one-electron densities of states (DOS) calculated for the BM-SFO, DD-SFO$_{2.75}$ with in-plane ($\parallel$) and out-of-plane ($\perp$) FeO$_5$ pyramidal connectivity, ID-SFO$_{2.75}$ with occupied out-of-plane O sites, and P-SFO. In the case of BM-SFO ($x = 2.5$), both $\parallel$ and $\perp$ OVC configurations show a band gap of over 1 eV and nearly identical band edge DOS profiles. The band gap is underestimated with respect to the experimental values of $\approx 2.0$ eV[21], as expected for PBEsol. For $x = 2.75$, intercalation type diffusion (ID-SFO$_{2.75}$) results in the band gap closure, indicating a metallic behavior. In contrast, bulk-phase DD-SFO$_{2.75}$ formed by oxidizing in-plane OVC remain nonmetallic regardless of the orientation of the remaining V$_O$ channels. As the oxygen content increases to $x = 3.0$, the DOS magnitude near the Fermi level increases continuously indicating enhanced metallic conductivity. The band gap closure upon increasing oxygen content from $x = 2.5$ to $x = 3.0$ is driven by the appearance of unoccupied Fe 3$d$ states (shown with shaded areas in Fig. 4e, f), which are depleted by the incorporation of O species into the OVCs. Thus, the electronic properties of the SrFeO$_x$ films, particularly the onset for insulator-to-metal transition, can be controlled not only by changing the oxygen content but also by controlling the oxidation pathways to access the appropriate precursor phases.

In summary, we demonstrated the ability to promote and guide the transfer of oxygen species from reducible substrates to oxidize differently oriented BM-SFO thin films and monitor the structural changes and phase transitions using in-situ TEM. By combining precisely controlled synthesis of precursor SFO with in-situ control of the electron-beam-induced electric field and sample heating, we were able to activate oxygen diffusion along selected pathways in BM-SFO that allows us to access two different reaction intermediates – Disruptive Diffusion-SFO$_{2.75}$ and Intercalation Diffusion-SFO$_{2.75}$ phases before the films were fully oxidized to metallic P-SFO. For the case of OVCs parallel to the interface, O diffusion is found to involve both FeO$_6$ octahedral and FeO$_4$ tetrahedral sublayers, whereas the phase transition from BM-SFO to P-SFO proceeds through a DD-SFO$_{2.75}$ intermediate state via a disruptive diffusion process. In contrast, intercalation-only O diffusion is realized in samples displaying OVCs perpendicular to the interface, which results in a meta-stable ID-SFO$_{2.75}$ phase containing alternating FeO$_6$ octahedral and FeO$_5$ pyramidal sublayers. Our DFT calculations predict that differences in crystal field associated with these two SrFeO$_{2.75}$ intermediates result in qualitatively different electronic structures. It can be envisioned that selectively activating intercalation type diffusion may lead to faster ion transport, lower power assumption, and metastable intermediate with unique properties that can be harnessed for energy and information storage applications.

## Methods

### Thin film preparation
Epitaxial SrFeO$_{2.5}$ thin films with the thicknesses of 15−35 nm were grown on (001)-oriented SrTiO$_3$ and LSAT single crystal substrates using PLD[19]. The laser pulse (248 nm) energy density was $\approx 2$ J cm$^{-2}$, the repetition rate was 1 Hz. The substrates were heated to 700 °C during deposition and the growth oxygen pressure was kept at 0.1 mTorr. After growth, the samples were cooled down to room temperature under the same oxygen pressure.

### In-situ TEM experiments
The TEM samples used for in-situ TEM were prepared using a focus ion beam scanning electron microscopy (Helios). In-situ TEM experiments were conducted using an FEI Titan 80-300 TEM equipped with an aberration corrector for the objective lens and a Gatan furnace-based heating holder. The accelerate voltage of 300 kV and electron beam dose rate of $\approx 10^3$ e Å$^{-2}$ s$^{-1}$ were used in the in-situ TEM experiments. The TEM samples were heated to elevated temperatures (200 to 300 °C) during the experiments to promote the reaction, making it suitable for in-situ TEM observation. The high angle annular dark-field (HAADF) scanning transmission electron microscopy (STEM) image, annular bright-field (ABF) STEM image and electron energy-loss spectrum (EELS) mapping were conducted using JEM ARM200F. The collection angle for HAADF and ABF imaging were 90-370 mrad and 10−23 mrad, respectively. The probe current of $\approx 20$ pA was used for STEM imaging and EELS mapping to minimize the electron beam induced phase transition. The Dual-EELS was used for the energy calibration of Fe-L edge with the simultaneously acquired zero loss spectrum.

### SIMS measurements
ToF-SIMS measurements were performed using a ToF-SIMS V (ION-TOF GmbH, Münster, Germany) mass spectrometer equipped with a time-of-flight analyzer of a reflectron type. A dual-beam non-interlaced depth profiling strategy was used, in which a 1.0 keV Cs$^+$ beam (~40 nA, 200 μm × 200 μm scanning area) was used for sputtering and a 50 keV Bi$_3^{2+}$ beam (~0.05 pA, 50 μm × 50 μm scanning area at the Cs$^+$ crater center) was used for negative spectra data collection. One BM-SFO sample was first annealed in vacuum at 700 °C for 0.5 h to promote further oxygen loss in SFO. After vacuum annealing, the SFO/STO sample, together with an untreated STO(001) substrate (Reference STO), were annealed at 650 °C for 0.5 h in a tube furnace backfilled with 50 Torr of $^{18}$O$_2$ (97% purity, Cambridge Isotopes). The SFO/STO interface location was confirmed by the secondary ion signals of FeO$^-$ and TiO$^-$.

### Ab-initio simulations
The calculations were performed using the Vienna Ab-initio Software Package (VASP)[57,58]. The Perdew-Burke-Ernzerhof exchange-correlation functional modified for solids (PBEsol)[59] and the projector augmented wave pseudopotentials[60], as implemented in VASP, were used. The energy cut-off was 500 eV. SFO was represented using the periodic model approach and a supercell equivalent to the 4 × 4 × 4 extension of the pseudo-cubic perovskite crystallographic cell. Gamma point only was used for energy minimization with respect to the internal coordinates and the out-of-plane lattice parameter, the electronic structure for the energy minimum configuration was recalculated using 2 × 2 × 2 Monkhorst-Pack k-points mesh, 4 × 4 × 4 mesh for used for DOS calculations. The total energy was converged to 10$^{-5}$ eV. The Hubbard $U$ correction for Fe 3$d$ states ($U_{eff} = U − J = 3$ eV) was applied using Dudarev's approach[61]. The 1 × 2 × 1 k-mesh and $U_{eff} = 0$ eV were used for the 8 × 2 × 4 supercell. Atomic charges were calculated using the Bader's approach[62]. The diffusion pathways and activation energies were calculated for the SrTiO$_3$ substrate ($a = b = 3.905$ Å) using the nudged elastic band (NEB) method and eight NEB images unless stated otherwise. Energy gain due to oxygen incorporation was calculated with respect to the gas-phase O$_2$ molecule.

### Reporting summary
Further information on research design is available in the Nature Portfolio Reporting Summary linked to this article.

## Data availability
The data that support the findings of this study are available from the corresponding authors upon request.

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

## Acknowledgements

The in-situ TEM work was supported by the U.S. Department of Energy (DOE), Office of Science (SC), Office of Basic Energy Sciences (BES), Early Career Research Program under award #68278. ToF-SIMS characterization and DFT simulations were supported by the DOE BES Division of Materials Sciences and Engineering under Award No. 10122. We thank Dr. Endong Jia for valuable discussions and help on the Figure layout. A portion of the research was performed using EMSL (Ringgold ID130367), a DOE User Facility sponsored by the Office of Biological and Environmental Research and located at the Pacific Northwest National Laboratory (PNNL). This research used resources of the National Energy Research Scientific Computing Center, a DOE SC User Facility supported by the SC of the DOE under Contract No. DE-AC02-05CH11231 using NERSC award BES-ERCAP0021800. PNNL is a multi-program national laboratory operated for DOE by Battelle under contract DE-AC05-76RL01830.

## Author contributions

Y.D. and Z.Y. designed and initiated the research. L.W. grew the thin film samples. Z.Y. and C.W. conducted the in-situ TEM experiments and performed data analysis. P.V.S. planned and performed the DFT calculations. M.B. performed XRD measurements. W.L., J.D., M. E., and Z.Z. performed the 18O annealing and SIMS experiment. All authors participated in the discussion and interpretation of results. Z.Y., L.W., S.A.C., P.V.S., and Y.D. wrote the manuscript.

## Competing interests

The authors declare no competing interests.
