## [Peer Review File · Nature Communications]

Guided anisotropic oxygen transport in vacancy ordered oxidesEditorial Note: This manuscript has been previously reviewed at another journal that is not operating a transparent peer review scheme. This document only contains reviewer comments and rebuttal letters for versions considered at *Nature Communications*.

REVIEWER COMMENTS

Reviewer #2 (Remarks to the Author):

In my first report, I suggested some revisions. The authors have adequately responded to these suggestions, and made the required revisions within the manuscript.

The data from the experiments is well presented and does, to my opinion, support the conclusions made in the manuscript. I propose to publish the manuscript in *Nature Communications*.

Reviewer #3 (Remarks to the Author):

The authors of this paper have made some changes to the manuscript in light of reviewers comments, and have incorporated a number of new references. Whilst this has addressed some of the concerns, there are others that remain.

It is troubling that the authors used one isotopic labelling experiment on one sample (sample set?) and having this result suggest that it supports all other conclusions. I would prefer to see conclusions supported by reproducible data, and not inferred from a single result.

I am also concerned with the STO 18O₂ anneals. As stated originally the untreated STO will be modified by the PLD process, and hence this is not a direct comparison. The issue here is that the reference state is a fully oxidised STO, not one subjected to annealing in vacuum. The text states that the BM-SFO/STO sample was vacuum annealed, whilst the STO reference was not. This is not a reasonable comparison. It should also be noted the the prior literature notes that to exchange STO with oxygen 18 took one week at 1000C. This calls into question the whole experiment. The claim that the BM-SFO draws oxygen from the

film, showing enhanced ^{18}O oxygen concentration would decrease the isotope in the STO substrate - I am also not convinced that the authors have fully accounted for matrix effects and sputter yields on moving from film to substrate. Have the authors determined the oxygen diffusion coefficient for STO, and does this agree with their analysis? The schematic in Fig 2h shows vacancy creation in SFO, but will this also drive vacancy creation in STO? This is implied but not proven. This also does not account for any oxygen diffusion process laterally at the interfaces, increasing the overall intensity of the signals. An 87% enrichment is remarkable and implies either a virtually fully reduced substrate or remarkably fast oxygen exchange kinetics. Both seem unlikely and the authors explanation is unconvincing.

I also find that the authors have omitted a further relevant reference regarding the formation of $\text{SrFeO}_{2.75}$ - A Maity et al 2015 J. Phys. D: Appl. Phys. 48 504004.

Overall the work reported has been improved, but there are still issues, particularly surround the analysis of the isotopic labelling experiments.

Reviewer 2

In my first report, I suggested some revisions. The authors have adequately responded to these suggestions, and made the required revisions within the manuscript.

The data from the experiments is well presented and does, to my opinion, support the conclusions made in the manuscript. I propose to publish the manuscript in Nature Communications.

***Our Response:** We thank the Reviewer for his/her positive comments on our manuscript.*

Reviewer 3

The authors of this paper have made some changes to the manuscript in light of reviewers comments, and have incorporated a number of new references. Whilst this has addressed some of the concerns, there are others that remain.

Our Response: *We appreciate the Reviewer for his/her detailed review of our manuscript. We have also revised our manuscript to address his/her comments/concerns below.*

Comment 1: It is troubling that the authors used one isotopic labelling experiment on one sample (sample set?) and having this result suggest that it supports all other conclusions. I would prefer to see conclusions supported by reproducible data, and not inferred from a single result.

Our Response: *According to the Reviewer's suggestion, we repeated the isotopic labeling experiment on two STO substrates (one is vacuum annealed and another one is non-vacuum annealed). We have also repeated the ToF-SIMS measurements, as shown in the Figures below. These results strongly support our conclusions in the previous version of the manuscript. In light of the Referee's feedback, we have rewritten the second paragraph on page 6, revised the Figure 2 in the main text and added Figures S5 to the SI file. We have also added two more authors (Jeffrey A. Dhas, Mark H. Engelhard) to the author list and three more literatures (Adv. Funct. Mater. 27, 1700243, 2017; J. Sens. Sens. Syst. 6, 107–119, 2017; Acta Materialia 58, 457–463, 2010) to the reference list (Ref. 47-49) in our revised manuscript.*

*“To further visualize the oxygen exchange between SFO and STO, we performed ^{18}O labeled, time-of-flight secondary-ion mass spectrometry (ToF-SIMS) analysis. One BM-SFO/STO sample (~15 nm thickness) and a STO(001) substrate were first annealed in vacuum at 700 °C for 1 hour. For BM-SFO/STO sample, vacuum annealing (VA) could promote oxygen loss in SFO, which would subsequently draw oxygen from the STO substrate. This is similar to the well-established phenomenon of oxygen transfer from the substrate to the film deposited by PLD^{32,33}. After VA, the BM-SFO/STO sample and the STO substrate (STO reference marked in **Fig. 2h**) were annealed at 650 °C for 0.5 h in a tube furnace backfilled with 50 Torr of $^{18}\text{O}_2$. The ToF-SIMS depth profiles of the ^{18}O enrichment level (c , defined as $^{18}\text{O}/(^{18}\text{O}+^{16}\text{O})$) for both samples are shown in **Fig. 2h**. The ^{18}O enrichment level is ~74% at the near-surface region of STO reference, which undergoes an exponential decay to natural abundance level (~0.2%) within a ~100 nm range (see **Supplementary Fig. S5**). In comparison, the ^{18}O enrichment level is higher (~80% in SFO film) in the BM-SFO/STO sample due to the strong surface oxygen exchange between $^{18}\text{O}_2$ and SFO. An even higher ^{18}O enrichment level (up to ~87%) is also observed on the STO side close to the STO/SFO interface, indicating that the oxygen vacancies in the STO substrate created during the VA process have been filled by $^{18}\text{O}_2$ annealing (see schematics shown in the top and middle of*

Fig. 2h). We also performed ^{18}O annealing experiment on a non-VA STO substrate and observed similar ^{18}O concentration distribution with that of VA STO substrate (see **Supplementary Fig. S5**). These results clearly demonstrated that oxygen exchange of the STO substrate can occur at 650°C . Moreover, the oxygen diffusion coefficient (D) of STO at 650°C was determined by fitting the ^{18}O depth profiles for both VA and non-VA STO substrates. As shown in **Supplementary Fig. S5**, D is on the order of $10^{-21} - 10^{-20} \text{ m}^2/\text{s}$, in agreement with that reported in previous work.⁴⁷⁻⁴⁹”

Figure 2 | Phase transformations of BM-SFO→SrFeO_{2.75}→P-SFO observed on BM-SFO/STO with starting OVCs // STO(001). **a-c**, Selected in situ TEM images (see Supplementary Movie S1) of the same region of the SFO sample illustrating stages of the phase transition process over ~300 s: the initial state of BM-SFO (**a**), appearance of the P-SFO inclusions (**b**), and propagation of the P-SFO phase (from the bottom of the image) (**c**). **d-f**, High resolution TEM images (top), together with the Fast Fourier Transform patterns (bottom left) and simulated diffraction patterns (bottom right) for (**d**) BM-SFO, (**e**) SrFeO_{2.75}, and (**f**) P-SFO. **g**, Structure model of bulk-like SrFeO_{2.75}. **h**, Schematics showing the creation of oxygen vacancies (V_{O}) in SFO/STO by annealing in vacuum (top), and subsequent replenishment of

oxygen by annealing in $^{18}\text{O}_2$ (middle). ToF-SIMS depth profiles (bottom) display the ^{18}O enrichment level for a 15 nm SFO film grown on STO(001) and a STO reference sample. The dashed line denotes the SFO/STO interface with SFO film on the left side.

Figure S5 | Determine the oxygen diffusion coefficient (D) of STO substrate. a, ToF-SIMS ^{18}O depth profiles for VA STO and non-VA STO substrates. The black dashed line denotes the ^{18}O experimental natural abundance level. **b,c,** Data fitting for (b) VA and (c) non-VA STO substrates, corrected by average measured ^{18}O background signal (black dashed line).

Prior to data fitting, ^{18}O data was normalized according to Equation 1. To fit the experimental ^{18}O depth profiles, the equation for transient diffusion in a semi-infinite medium was used (Equation 2). Here, c_1 is the concentration of ^{18}O in the annealing gas, c_0 is the average measured background, z is the depth, t is the ^{18}O tracer annealing time (1800 seconds), and D is the diffusion coefficient. Notably, surface exchange kinetics are ignored in this model^{5,6}. A diffusion coefficient range was established to capture the ^{18}O depth profile for both VA and non-VA STO substrates. The range is on the order of $10^{-21} - 10^{-20} \text{ m}^2/\text{s}$.

$$c = \frac{[^{18}\text{O}]}{[^{16}\text{O}] + [^{18}\text{O}]} \quad (1)$$

$$\frac{c(z, t) - c_0}{c_1 - c_0} = 1 - \text{erf} \left[\frac{z}{2\sqrt{Dt}} \right] \quad (2)$$

Comment 2: I am also concerned with the STO $^{18}\text{O}_2$ anneals. As stated originally the untreated STO will be modified by the PLD process, and hence this is not a direct comparison. The issue here is that the reference state is a fully oxidized STO, not one subjected to annealing in vacuum. The text states that the BM-SFO/STO sample was vacuum annealed, whilst the STO reference was not. This is not a reasonable comparison. It should also be noted the prior literature notes that to exchange STO with oxygen 18 took one week at 1000C. This calls into question the whole experiment.

Our Response: We thank the Reviewer for pointing this out. In response to the Reviewer's comment, we have performed additional $^{18}\text{O}_2$ annealing experiments (650°C, 0.5 h) and ToF-SIMS analysis on vacuum-annealed (VA) and non-vacuum-annealed (non-VA) STO substrates. We observed similar ^{18}O concentration distribution for VA and non-VA STO substrates (see our response to Comment 1). The extracted diffusion coefficient (D) is on the order of $10^{-21} - 10^{-20} \text{ m}^2/\text{s}$, in good agreement with that reported in previous work,⁴⁷⁻⁴⁹ demonstrating that oxygen exchange of the STO substrate can occur at 650°C. We also want to point out that in our previous work, we grew WO_3 on STO(001) at 500°C in $^{18}\text{O}_2$ (3×10^{-6} Torr) and observed clear oxygen exchange between $^{18}\text{O}_2$ and STO. [ACS Appl. Mater. Interfaces 10(20), 17480-17486, 2018]

Comment 3: The claim that the BM-SFO draws oxygen from the film, showing enhanced ^{18}O oxygen concentration would decrease the isotope in the STO substrate. - I am also not convinced that the authors have fully accounted for matrix effects and sputter yields on moving from film to substrate.

Our Response: We would like to clarify the oxygen exchange process. The oxygen extraction from STO side occurs during vacuum annealing, where oxygen released to vacuum through SFO contains ~0.2% ^{18}O (natural abundance). During annealing in $^{18}\text{O}_2$, the vacancies in SFO and reduced STO at the interface are replenished with ^{18}O . That's why there is a locally enhanced ^{18}O signal at the STO interface. (Fig. 2h).

ToF-SIMS isotopic analysis is hardly affected by matrix effect, because all isotopes of an element should have almost identical chemical properties. In SIMS isotopic analysis, signal intensity may play a more important role, because the detector may suffer from so-called saturation issue. In our ToF-SIMS testing, the primary ion beam current was carefully optimized to avoid any un-controllable saturation. We combined film thickness measured by cross-sectional TEM, elemental signals from ToF-SIMS, and crater depth measurement by profilometer after ToF-SIMS to calibrate the sputtering rate, and no obvious change in sputtering rate (~0.41 nm/s) was detected.

Comment 4: Have the authors determined the oxygen diffusion coefficient for STO, and does this agree with their analysis?

Our Response: We thank the Reviewer for this insightful question. In response to the Reviewer's comment, we have added the following sentences on page 6 in our revised manuscript.

*“Moreover, the oxygen diffusion coefficient (D) of STO at 650°C was determined by fitting the ^{18}O depth profiles for both VA and non-VA STO substrates. As shown in **Supplementary Fig. S5**, D is on the order of $10^{-21} - 10^{-20} \text{ m}^2/\text{s}$, which is comparable with that reported in previous studies.⁴⁷⁻⁴⁹”*

Comment 5: The schematic in Fig 2h shows vacancy creation in SFO, but will this also drive vacancy creation in STO? This is implied but not proven. This also does not account for any oxygen diffusion process laterally at the interfaces, increasing the overall intensity of the signals. An 87% enrichment is remarkable and implies either a virtually fully reduced substrate or remarkably fast oxygen exchange kinetics. Both seem unlikely and the authors explanation is unconvincing.

Our Response: *The schematic in Fig. 2h indeed shows vacancy creation in both SFO and STO. This is proven by the ToF-SIMS experiment where the STO close to the SFO interface shows high degree of ^{18}O incorporation. After our in situ TEM experiment, a visible decrease of O intensity for STO in the region close to the STO/SFO interface from TEM elemental composition profiles (shown in Figure S4) also verify that the oxygen vacancies created in the STO substrate. The high degree of ^{18}O enrichment at the interface STO is due to the fast oxygen exchange kinetics and oxygen conduction of SFO, as have been discussed previously. Materials systems containing ordered oxygen vacancy channels can facilitate $^{18}\text{O}_2$ oxygen surface exchange and diffusion in the bulk, leading to the enrichment of the substrates. For example, for $\text{La}_{1-x}\text{Sr}_x\text{CoO}_{3-\delta}$ thin films grown on STO, significant oxygen enrichment of STO was found after 400°C and 5 min $^{18}\text{O}_2$ annealing. [ACS Nano 2013, 7, 4, 3276–3286]*

We would like to further quantify the difference of surface exchange and diffusion kinetics in orientation controlled SFO systems in future studies, but the process has been delayed as we are in the process of moving our thin film growth lab to a new building at PNNL.

Comment 6: I also find that the authors have omitted a further relevant reference regarding the formation of $\text{SrFeO}_{2.75}$ - A Maity et al 2015 J. Phys. D: Appl. Phys. 48 504004.

Our Response: *We thank the Reviewer for bringing this excellent study to our attention. We have added this paper as a new reference (Ref. 44) in our revised manuscript.*

Comment 7: Overall the work reported has been improved, but there are still issues, particularly surround the analysis of the isotopic labelling experiments.

Our Response: *We thank the Reviewer again for his/her detailed comments on our manuscript. We hope he/she will be satisfied with the changes we have made in our revised manuscript.*

REVIEWER COMMENTS

Reviewer #3 (Remarks to the Author):

The authors have further improved their manuscript but concerns still remain.

The additional measurements are welcome, but these add further questions. Firstly it is not clear why the authors have been unable to present a fit of their diffusion data. This is typically easily achieved, however the authors choose to present estimates and a range of diffusion coefficient. Secondly they do not appear to have fully appreciated the data of de Souza (their ref 34) in which diffusion and defect chemistry in STO is discussed, including an initial 40 nm profile at the near surface due to space charge effects.

I am also further concerned that the authors have misunderstood the comment regarding matrix effects. The issue raised was that of moving from the SFO matrix into the STO matrix. The SFO matrix is different from the STO matrix and hence the chemical environment of the oxygen species is different, and so, as the ionisation of the species is affected by its surroundings, the normalised oxygen signal across the interface may be affected. Hence a change in oxygen intensity on moving across an interface could be due to the different ionisation yields due to the chemical differences between the materials. The authors do not appear to have considered this (or worse, erroneously dismissed it!). They appear to suggest that there are no matrix effects in TOFSIMS. This is incorrect. For the same species in the same matrix (i.e. oxygen isotopes in STO) this is correct, but as soon as the matrix changes this assumption is no longer valid. For a fuller discussion of matrix effects the authors are recommended to look at Priebe et al, *J. Anal. At. Spectrom.*, 2020,35, 1156-1166, in which metal alloys are discussed, and Isa et al, <https://doi.org/10.1016/j.chemgeo.2017.03.020>. Further the following these also discusses these effects <https://theses.hal.science/tel-00721832>. Finally the paper by De Souza and Martin is recommended (DOI 10.1002/pssc.200675227).

I am also perplexed on looking at the TEM data in Fig 2a. The authors claim to have a 15nm film of BM-SFO on STO. Given that the scale bar is 5 nm, this suggests that the film is actually 35 - 40 nm in thickness. This would shift the position of the interface in the SIMS

data closer to the increase in signal. It is also noticeable that the authors state that the interface location was confirmed with the Ti and Fe signals, but these are not presented even as SI. And which signals were being probed, and was this simultaneous i.e. were both +ve and -ve spectra collected? Was the species collected the $^{16}\text{O}/^{18}\text{O}$ - and if so presumably for the metal cations the preferred species would be the FeO^+ , TiO^+ which require a secondary experiment?

Further revisions are, in this reviewer's opinion, required.

Reviewer #3

The authors have further improved their manuscript but concerns still remain.

***Our Response:** We appreciate the Reviewer for recognizing the improvements we have made based on his/her previous feedback; we address the additional concerns in our response provided below.*

Reviewer Comment 1: The additional measurements are welcome, but these add further questions. Firstly it is not clear why the authors have been unable to present a fit of their diffusion data. This is typically easily achieved, however the authors choose to present estimates and a range of diffusion coefficient. Secondly they do not appear to have fully appreciated the data of de Souza (their ref 34) in which diffusion and defect chemistry in STO is discussed, including an initial 40 nm profile at the near surface due to space charge effects.

***Our Response:** We thank the Reviewer for highlighting the relevance of De Souza's work on the diffusion and defect chemistry in STO (ref 34 in the previous version of manuscript) to our study. Previously we utilized the relatively simple diffusion through a semi-infinite medium model, which was unable to provide a proper fit of the oxygen tracer profile within STO. We have thoroughly reviewed the related work and recognize its significance in understanding diffusion mechanisms and space charge effects in oxide materials. We have employed the diffusion models described in De Souza's work and successfully fitted our diffusion data in COMSOL, as shown in the following Figure 1b and 1c. Reviewer's insights have been instrumental in further clarifying this aspect of our study.*

*In light of Reviewer's comments, we have revised Figure S6 in SI and added the following sentences to page 6 of our revised manuscript: "Moreover, the oxygen diffusion coefficient of STO at 650°C was determined by fitting the ¹⁸O depth profiles obtained from both VA and non-VA STO substrates^{34,51-54}. As shown in **Supplementary Fig. S6**, the bulk diffusion coefficients (D^*) for VA and non-VA STO substrates were found to be 1.2×10^{-10} and 3.1×10^{-11} cm²/s, respectively, in agreement with values previously reported in the literature^{51,52}."*

Figure 1 (Figure S6 in SI) | Determination of the bulk oxygen diffusion coefficient (D^*) in STO. a, ToF-SIMS ^{18}O - depth profiles for VA STO and non-VA STO substrates. The black dashed line denotes the ^{18}O experimental natural abundance level. **b,c,** Data fitting for (b) non-VA and (c) VA STO substrates, where k^* is the surface exchange coefficient and Φ_0 is the space-charge potential.

Reviewer Comment 2: I am also further concerned that the authors have misunderstood the comment regarding matrix effects. The issue raised was that of moving from the SFO matrix into the STO matrix. The SFO matrix is different from the STO matrix and hence the chemical environment of the oxygen species is different, and so, as the ionisation of the species is affected by its surroundings, the normalised oxygen signal across the interface may be affected. Hence a change in oxygen intensity on moving across an interface could be due to the different ionisation yields due to the chemical differences between the materials. The authors do not appear to have considered this (or worse, erroneously dismissed it!). They appear to suggest that there are no matrix effects in TOFSIMS. This is incorrect. For the same species in the same matrix (i.e. oxygen isotopes in STO) this is correct, but as soon as the matrix changes this assumption is no longer valid. For a fuller discussion of matrix effects the authors are recommended to look at Priebe et al, J. Anal. At. Spectrom., 2020,35, 1156-1166, in which metal alloys are discussed, and Isa et al, <https://doi.org/10.1016/j.chemgeo.2017.03.020>. Further the following these also discusses these effects <https://theses.hal.science/tel-00721832>. Finally the paper by De Souza and Martin is recommended (DOI 10.1002/pssc.200675227).

Our Response: We now see that our previous response did not sufficiently address the matrix effect concern raised by the Reviewer. We appreciate the Reviewer for providing additional references for further understanding and the opportunity to address these issues. We have thoroughly reviewed the recommended references, including Priebe et al. (J. Anal. At. Spectrom., 2020, 35, 1156-1166), Isa et al. (Chemical Geology, 2017, 458, 14-21), the thesis by Matthieu Py, and the paper by De Souza and Martin (Phys. Status Solidi C 2007, 4, 1785– 1801). These sources provide valuable insights into matrix effects,

particularly in relation to metal alloys and different chemical compositions. We greatly appreciate the Reviewer's guidance in pointing us towards these excellent references, and performed additional experiments to evaluate the matrix effect. We have added the following discussion to page 6 of our revised manuscript:

“To account for the possibility of matrix effects during ToF-SIMS analysis, one non- $^{18}\text{O}_2$ annealed BM-SFO/STO sample was measured (Fig. S5). The oxygen signal intensity shows a notable increase upon sputtering through the BM-SFO film into the STO substrate (Fig. S5b). Using the ratio of the total oxygen signal intensity on the film and substrate sides (i.e., $(^{16}\text{O}+^{18}\text{O})_{\text{film}}/(^{16}\text{O}+^{18}\text{O})_{\text{substrate}}$) yields a ratio of 0.80, which is close to the expected stoichiometric ratio of oxygen in BM-SrFeO $_{2.5}$ /SrTiO $_3$ of 0.83. Therefore, this result suggests that the matrix effect, which has been well described in previous reports,⁴⁷⁻⁵⁰ does not play a dominant role in the oxygen signal intensity change upon sputtering through the BM-SFO film into the STO substrate.”

“Based on prior reports,⁴⁷⁻⁵⁰ matrix effects can significantly influence signal intensity, but play a minimal role in isotopic ratio counting. Indeed, as demonstrated in Fig. S5, the ratio of $^{18}\text{O}/(^{16}\text{O}+^{18}\text{O})$ for a non- $^{18}\text{O}_2$ annealed BM-SFO/STO remains constant at the natural abundance of ^{18}O (Fig. S5c), even though the signal intensity changes from BM-SFO to STO. Hence, we utilized the ^{18}O enrichment level, defined as $n^* = ^{18}\text{O}/(^{18}\text{O}+^{16}\text{O})$, to assess and visualize the oxygen exchange occurring at the interface. Figure 2h presents the depth profiles obtained from ToF-SIMS, illustrating the ^{18}O enrichment level for both $^{18}\text{O}_2$ annealed samples.” In addition, we included Figure 2 (as Figure S5) to the SI file. We have also included the four reports (cited above) in the reference list in our revised manuscript.

Figure 2 (Figure S5 in SI) | Examining the influence of potential matrix effects on interface oxygen intensity of a non- $^{18}\text{O}_2$ annealed BM-SFO(~15 nm)/STO. a, The interface location of SFO/STO was confirmed by the secondary ion signals of FeO- and TiO-. b, Depth profiles of the ^{16}O - and ^{18}O - signals obtained via ToF-SIMS. c, Depth profiles of the total oxygen signal (purple) and normalized ^{18}O - signal (green). The vertical grey lines denote the interface.

Reviewer Comment 3a: I am also perplexed on looking at the TEM data in Fig 2a. The authors claim to have a 15nm film of BM-SFO on STO. Given that the scale bar is 5 nm, this suggests that the film is actually 35 - 40 nm in thickness. This would shift the position of the interface in the SIMS data closer to the increase in signal.

Our Response: *We appreciate the Reviewer for bringing this discrepancy to our attention, and we apologize for any confusion caused. We want to clarify that the thickness of the SFO sample used for ToF-SIMS measurements is different from the one used for in situ TEM studies. In our revised manuscript, we have explicitly mentioned that "Epitaxial BM-SFO thin films with the thicknesses of 15-35 nm were deposited on (001)-oriented SrTiO₃ (STO) and (LaAlO₃)_{0.3}(Sr₂AlTaO₆)_{0.7} (LSAT) single crystal substrates by using pulsed laser deposition (PLD)" in both the Results section (page 3) and the Methods section (page 11).*

To address the Reviewer's concern and provide further clarity, we have made the necessary updates to the caption of Figure 2 in the main text of the manuscript (shown as Figure 3 in this letter). Specifically, we have added the thickness information of the SFO films used for both in situ TEM studies and ToF-SIMS measurements.

Reviewer Comment 3b: It is also noticeable that the authors state that the interface location was confirmed with the Ti and Fe signals, but these are not presented even as SI. And which signals were being probed, and was this simultaneous i.e were both +ve and -ve spectra collected? Was the species collected the 16O/18O- and if so presumably for the metal cations the preferred species would be the FeO+, TiO+ which require a secondary experiment?

Our Response:

*We agree with the Reviewer that adding extra information on the Fe and Ti signals will be beneficial, and we have added it to Fig.2h (Fig.3 in this letter, as shown below) in the main text and Fig.S5 (Fig.2 in this letter) in the SI section, respectively. And we have included the sentence, "The SFO/STO interface location, marked by the black dashed line, was confirmed by the secondary ion signals of FeO- and TiO-" in the caption to clarify the confirmation method for the SFO/STO interface location. Because the spectrum was measured in the negative mode during the ToF-SIMS analysis, only the negative signals, including the ¹⁶O-, ¹⁸O-, FeO-, and TiO-, were probed. Furthermore, the FeO- and TiO- signals recorded during negative spectral data collection could also be used to determine the position of the film/substrate contact. To clarify, we revised the SIMS methods section in the main manuscript (Page 12): "A dual-beam *non-interlaced* depth profiling strategy was used, in which a 1.0 keV Cs⁺ beam (~40 nA, 200 μm × 200 μm scanning area) was used for sputtering and a 50 keV Bi₃²⁺ beam (~0.05 pA, 50 μm × 50 μm*

scanning area at the Cs^+ crater center) was used for *negative spectra* data collection. ... *The SFO/STO interface location was confirmed by the secondary ion signals of FeO- and TiO-.*”

Figure 3 (Figure 2 in the revised manuscript) | Phase transformations of *BM-SFO*→*SrFeO*_{2.75}→*P-SFO* observed on *BM-SFO/STO* with starting OVCs // *STO(001)*. a-c, Selected *in situ* TEM images (see Supplementary Movie S1) of the same region of a ~35 nm SFO film illustrating stages of the phase transition process over ~300 s: the initial state of *BM-SFO* (a), appearance of the *P-SFO* inclusions (b), and propagation of the *P-SFO* phase (from the bottom of the image) (c). d-f, High resolution TEM images (top), together with the Fast Fourier Transform patterns (bottom left) and simulated diffraction patterns (bottom right) for (d) *BM-SFO*, (e) *SrFeO*_{2.75}, and (f) *P-SFO*. g, Structure model of bulk-like *SrFeO*_{2.75}. h, Schematics showing the creation of oxygen vacancies (V_{O_s}) in *SFO/STO* by annealing in vacuum (top), and subsequent replenishment of oxygen by annealing in ¹⁸O₂ (middle). ToF-SIMS depth profiles of ¹⁸O (bottom) display the enrichment level for a 15 nm SFO film grown on *STO(001)* and a *STO* reference sample. The *SFO/STO* interface location, marked by the black dashed line, was confirmed by the secondary ion signals of FeO- and TiO-.

REVIEWERS' COMMENTS

Reviewer #3 (Remarks to the Author):

The authors have made significant revisions to the paper and it now addresses all of the points raised by the reviewer. It can now be recommended for acceptance.

Reviewer #3

The authors have made significant revisions to the paper and it now addresses all of the points raised by the reviewer. It can now be recommended for acceptance

***Our Response:** We thank the Reviewer for his/her positive comments on our manuscript.*